# Automatic Identification of Local Features Representing Image Content with the Use of Convolutional Neural Networks

**Paweł Tarasiuk**, **Arkadiusz Tomczyk *** and **Bartłomiej Stasiak**

Institute of Information Technology, Lodz University of Technology, ul. Wolczanska 215, 90-924 Lodz, Poland; pawel.tarasiuk@p.lodz.pl (P.T.); bartlomiej.stasiak@p.lodz.pl (B.S.)

**\*** Correspondence: arkadiusz.tomczyk@p.lodz.pl; Tel.: +48-42-631-39-57

**Abstract:** Image analysis has many practical applications and proper representation of image content is its crucial element. In this work, a novel type of representation is proposed where an image is reduced to a set of highly sparse matrices. Equivalently, it can be viewed as a set of local features of different types, as precise coordinates of detected keypoints are given. Additionally, every keypoint has a value expressing feature intensity at a given location. These features are extracted from a dedicated convolutional neural network autoencoder. This kind of representation has many advantages. First of all, local features are not manually designed but are automatically trained for a given class of images. Second, as they are trained in a network that restores its input on the output, they may be expected to minimize information loss. Consequently, they can be used to solve similar tasks replacing original images; such an ability was illustrated with image classification task. Third, the generated features, although automatically synthesized, are relatively easy to interpret. Taking a decoder part of our network, one can easily generate a visual building block connected with a specific feature. As the proposed method is entirely new, a detailed analysis of its properties for a relatively simple data set was conducted and is described in this work. Moreover, to present the quality of trained features, it is compared with results of convolutional neural networks having a similar working principle (sparse coding).

**Keywords:** image representation; local features; autoencoder; convolutional neural network; machine learning

## 1. Introduction

Images are typically represented using regular grids of pixels. The information about image content is kept both in pixels' attributes (color channels) and, which seems to be even more important, in their spatial distribution. This kind of representation, although natural for humans, has at least one crucial drawback: It significantly complicates the design of effective computer algorithms able to accomplish tasks which are relatively easy for our visual system. The nature of this problem lies not only in the huge number of image elements and the variety of their possible distributions, but also in the fact that humans do not consciously operate on individual pixels. In most of the cases, the latter reason makes it impossible to directly write computer programs imitating the unconscious process of image understanding.

There are two typical approaches allowing to overcome the above problem. The first group of methods aims at changing and simplifying the representation of image content. The second one, instead of direct implementation, engages machine learning for this purpose. Although these methods can be used separately (simplified representations may allow to design algorithms ready for direct

implementation and there are trainable models that can operate on raw pixels), they are usually combined together. The need for this combination originates in the limitations of existing models (e.g., specific input format) as well as in the necessity of selection of optimal model parameters (even carefully designed algorithms require problem specific fine-tuning).

An algorithm solving a given problem, created both with the use of machine learning techniques and without them, requires domain knowledge either to encode it in a computer program or to design the training objective. Unfortunately, domain experts usually have problems with sharing their knowledge in a form of ready-to-use mathematical formulas. They prefer to express it in imprecise natural language (it must be somehow adapted to become applicable in the source code of the program) or they provide it in a form of a training set (for a given input they specify the expected output). In both cases, it may be easier for them to operate on simplified representations rather than on millions of pixels. Training set preparation may be especially troublesome for some specific tasks. In particular, when precise image segmentation is the goal of analysis, the knowledge acquisition at pixel level can be tiresome and time-consuming. Moreover, if it concerns, e.g., medical applications, where the number of experts is limited, acquiring a huge and representative set of examples, required by most of machine learning algorithms, becomes almost impossible.

Another important aspect of image representation choice is interpretability of the algorithm results. Nowadays many artificial intelligence techniques have become a part of our life. Some of them are, or in the near future, will become responsible for our health and even life. Consequently, their authors must be prepared to at least explain their general principles to potential consumers to convince them that their product is safe. Operating on individual pixels makes it practically impossible. If instead the applied representation is significantly reduced and its components can be assigned a meaningful interpretation, such an explanation becomes plausible.

To conclude, the search for alternative image representations constitutes an important task from image analysis point of view. Moreover, it is very interesting itself as it may also allow better understanding of the principles of human visual system operation. There are many evidences that this system also tries to organize the recognition process creating several intermediate representations corresponding to elements of the observed scene [1]. Such intermediate representations are also observed in trained convolutional neural networks (CNN) [2–6] as they try to imitate the activity of visual cortex (to some extent). This was the main reason behind the choice of a CNN to automate the process of image representation construction in our research.

In this paper, we propose a new CNN-based method allowing to generate general image content representation. The image is described as a tuple of feature maps that describe the localization and intensity of selected visual features on the image plane. A feature map is a matrix that describes the intensity of a certain visual feature at every point of the image. This means that instead of global image features, such as *image is mostly blue* or *there are many vertical lines*, we focus on local features, such as *there is a vertical segment centered at a point with given coordinates*. Due to the local nature of the selected features, the proposed method encodes each feature occurrence as a single matrix element. The exact coordinates of the selected pixel reflect the precise localization of the feature in the image plane. As a result, the relation between feature maps and actual image content is much more direct than in the case of classic CNNs. This greatly simplifies the semantic analysis of features and feature maps, which originally required a specific approach such as deconvolutional neural networks [3].

Naturally, the aforementioned features have to be non-trivial. If visual features consisting of a single pixel were allowed, the trained feature extractor could take the form of an identity function and the feature maps would correspond to the original image. In order to select semantically meaningful features, our method ensures that the feature maps at a selected level are sparse matrices, where the neighborhood of each activated (non-zero) element is zeroed. The resulting image representation, based on the visual features of the image, should contain sufficient information to ensure that the goals of the analysis are met. The encoding is obviously lossy, but the most common patterns found in the training set are expected to be preserved. Because of its ubiquity in the field of machine learning, the MNIST

data set of handwritten digits [7] was employed for the purpose of the present study. Not only does it allow for comparisons of the obtained results with those produced by similar techniques, but it is also, due to its simplicity, easily interpretable. The latter asset enables understanding of the meaning of the generated local features.

The remaining part of this paper is organized as follows. Section 2 discusses related works. In Section 3, the concept of convolutional neural networks is outlined and our novel contributions are defined. Section 4 describes the experimental framework for providing as sparse an image representation as possible, without losing the key information. The results are discussed in Section 5. A detailed analysis of the neural network models is illustrated with various visualizations. Finally, Section 6 presents the conclusions and directions for further research.

## 2. Related Works

The method of local feature identification proposed in this work uses the specific properties of CNNs. To enforce sparse coding, which allows us to determine the precise localization of these features, we propose a specific neural network architecture with additional filtering layers and a unique adjustment of the training objective. This is a novel approach, and thus it is hard to compare it with existing works. Nevertheless, in this section we try to present some of the related works aggregating them into three groups: works devoted to other methods of generating alternative representations of image content; works trying to automatically find semantic interpretation of features emerging in CNNs; and works having, to some extent, similar working principles to our approach.

### 2.1. Image Representation

As it was mentioned in Section 1, the change of image representation (extraction of features) is a crucial step in image analysis. It depends naturally on the type of considered task and consequently on the techniques that will be used. In the case of image classification tasks, global representations may be sufficient. However, for pattern localization, object detection and, in particular, for image segmentation, local features extraction is essential. Global representations treat the image as a whole and try to generate descriptors (usually feature vectors) which summarize colors, textures, shapes, etc. visible in this image. Features presented in this work are local. It means that the descriptors are assigned not to the whole image, but to specific locations (keypoints) within it. Naturally, raw pixels are also such local descriptors, but what we look for are reduced representations where the number of descriptors is significantly smaller than the number of all pixels. The reduced representation does not necessarily mean the loss of information. Their number is smaller but as they describe properties of image regions they may contain more information than color channel values assigned to single pixels. Moreover, additional information may be also kept in the data structure reflecting relations (including spatial relations) between these descriptors.

In the literature, there are two typical strategies for local descriptor finding: The first one uses segmentation techniques to define homogeneous regions of the image. Having found them, descriptors may be assigned either to these regions [8] or to their borders [9]. The regions are associated with information about their precise location (e.g., centroid of the region) and, consequently, their spatial relations can be discovered as well. The second strategy achieves a similar goal in the opposite way. First, characteristic points are sought for in the image plane (keypoints) and the local region around them is identified afterwards. In this group, such techniques as SIFT [10] or SURF [11] can be mentioned. They are particularly interesting, as they provide scale and orientation invariance. In all the above cases, after region or keypoint detection, descriptors (local features) must be computed. These descriptors can take into account the shape and the color of a region or they can be based on local gradients (SIFT) or wavelet responses (SURF). All of them, however, are designed manually by the author of a specific application.

The local features can be used both to classify image content and to solve more complex tasks. In the simplest case, clustered descriptors allow the identification of visual vocabulary depending

on which bag-of-visual-words (BoVW) technique can be applied. In this approach, image content is transformed into a real vector (one-hot or frequency encoding) and consequently most classic pattern recognition techniques can be employed. If spatial relations between local features need to be taken into account or more complex tasks (object localization, segmentation) are to be solved, other methods must be used. For SIFT and SURF descriptors, a dedicated efficient matching algorithm was designed to find a correspondence of local features extracted from different images [10]. Other possible approaches construct a graph describing the image content, where local features are related to its nodes and spatial relationships are reflected in the edges. In such a case, geometric deep learning (GDL), allowing to generalize the CNN concept to non-Euclidean domains, can be applied [12–14]. Alternatively, active partitions [15], an extension of classic active contours, can be of use here as well.

### 2.2. Semantic Interpretation

There are many evidences that feature maps generated by successive convolutional layers of a CNN correspond with some semantically important parts of the analyzed images [3]. The identification of relationships between these parts and feature maps is not, however, a simple task. First of all, CNNs were always treated as trainable black boxes (similarly to other neural networks) and while designing their architecture no attention was paid to how the intermediate outputs can be interpreted. The resulting feature maps are usually blurred and it is really hard to understand the relation between them and the content of the input images. Moreover, in classic CNN architectures (pooling layers and no padding) the size of the feature maps is reduced in consecutive layers. This leads to further problems with identifying the precise location of semantically important regions.

In the literature there can be found several techniques trying to reveal the aforementioned relationships. In [4], first the peaks of the feature maps are mapped onto visual (receptive) fields within the input image. There their correspondence with known semantic parts is checked. As more than one feature map may be connected with a given part, a genetic algorithm is then applied to find the most appropriate subset of the feature maps from all the convolutional layers. In [5], instead of the layer outputs, their gradients maps, calculated with backpropagation algorithm, are used to find activation centers. In [16], the authors introduce class activation mapping (CAM), which can be used for identification and visualization of discriminative image regions, as well as for weakly supervised object localization. In that approach, a CNN network must be trained to classify images (supervised learning) and typical fully-connected layers are replaced by global average pooling (GAP) followed by a fully connected soft-max layer. The CAM for a given class can be found as a linear combination of the final feature maps generated by convolutional layers with weights corresponding to a specific network output. As the size of the class activation map is equal to the size of the final feature maps, it must be upsampled to be comparable with the input image. Finally, in [6], the authors assume that the top layers of a network correspond to the bigger parts or whole objects, while the lower layers reflect smaller parts which are building blocks for the more complex ones. They propose a method that is able to automatically discover a graph describing these relationships. It should be noted, that all these methods, although interesting, are quite complex. Moreover, they try to find correspondence with known parts (supervised process), which need not be optimal in every application.

### 2.3. Working Principles

The key part of the proposed method involves using sparse matrices as an intermediate step of image processing with CNNs. This should not be confused with sparse convolutional neural networks proposed in [17], as that work was founded upon using sparse filter matrices in multiple convolutional layers, whereas our approach is based on sparse outputs. Another method that applied sparse coding to CNNs was presented in [18] and addressed the problem of image super-resolution. This approach, however, also differs from ours in terms of both the main goal and the motivation behind using sparse matrices. In the present study, sparse matrices are utilized to generate image descriptions based on visual features.

These examples show that it is hard to find works with objectives similar to ours. Nevertheless, we were able to identify two groups of research areas which can be considered related and which will be used as a comparison base for our results.

Sparse coding in feed-forward neural networks was considered in multiple works as a tool for improving the performance of typical CNNs with dense matrices. This includes both theoretical analysis [19] and practical application to image reconstruction [20]. There are also multiple ways to generate sparse image representations for the tasks of image reconstruction and classification. One of the notable approaches is based on the Fisher discrimination criterion [21]. Multiple related works describe solutions based on CNNs [22–24], which make them similar and potentially comparable to the method presented in this paper. However, these studies are concerned with issues related to either the computational speed or accuracy and did not consider the problem of intermediate feature extraction. As presented in [25], sparse coding can simplify the classification task by maximizing the margin from the decision boundary in a selected metric space. It must be emphasized, however, that none of these works was dedicated to automatic detection of visual image features.

Sparse representation of the hidden layer outputs is also typical for spiking convolutional neural networks (SCNNs) [17]. The key component of SCNNs intends to simulate the electro-physiological process that occurs in synapses. Another notable advantage of SCNNs is the possibility to implement them on FPGA-based hardware [26]. The actual solutions are usually based on leaky integrate-and-fire (LIF) neurons [27,28] or spike-timing-dependent plasticity (STDP) learning [29]. Both above-mentioned methods involve the introduction of additional types of neurons that simulate spiking of the electrical charge, according to the selected model of synapses behavior. Distinct peaks related to the presence of specific patterns are rare enough to generate sparse data, which can be further reshaped into a sparse matrix. Thus, SCNN-based methods belong to the field of sparse coding. In the proposed approach, the learning process known from the basic CNNs is enhanced only with additional cost function components (which can be considered as model regularization) and activation functions, but no additional neuron types are introduced.

### 2.4. Contribution

The goal of our research was to create a tool that will be able to automatically (in an unsupervised way) discover spatially located visual features for a given class of images. These features should lead to a reduced representation of the image content without the loss of information contained there. Such a representation should allow to create image analysis algorithms which would be easier for interpretation, allowing the use of simpler models where external expert knowledge can be incorporated in a more natural way.

The image representation proposed in this work enables to achieve the above goal. Unlike the case of SIFT or SURF methods mentioned in Section 2.1, the feature identification does not rely on a manually designed algorithm, but it can be trained for a specific class of images. The role of a keypoint extractor is played by an encoder part of the proposed convolutional autoencoder. The value assigned to a given keypoint corresponds with the intensity of a visual feature. This value, together with the number of the sparse feature map where the keypoint was found, constitutes a form of a keypoint descriptor. It need not be more complex as no further matching is required when two images are compared. The feature map number directly identifies keypoints of the same type.

Although the types of the features (the numbers of the successive feature maps) seem to be very abstract, our approach allows us to discover and understand the nature of these features without the necessity of using such complex algorithms as those presented in Section 2.2. Their form can be revealed using a decoder part of our network. All of that would not be possible if we could not precisely locate these features in the original images, which is problematic for typical, blurred feature maps. Our approach solves this problem thanks to the novel training objective component, which enforces leptokurtic distribution of specific layer outputs, and thanks to the new filtering step added to the network architecture.

It should also be emphasized that the proposed representation differs significantly from reduced representations which can be obtained using classic feature reduction algorithms like PCA or non-convolutional autoencoder. The convolutional autoencoder used in this work takes the spatial relationships between reduced features and (which is its additional advantage) it performs feature reduction in a local way. The resulting features are calculated only on the basis of the pixels belonging to the respective receptive field. Such a weight sharing, typical for a CNN, reduces the number of trainable parameters and allows us to detect the same features in different places of the image plane regardless of the image size.

Finally, we use matrices, and not vectors, to encode images, because not only do we want to compress the information, but we want to extract the information about localization of the visual building blocks as well. This is a very specific application, which requires matrices only because the images are represented as regular grids of pixels. Nevertheless, the proposed approach is general. CNN is designed to work with matrix-like structures, but interpretation of these structures is irrelevant. If an autoencoder is used, one can obtain an encoder which generates sparse matrices preserving the whole information about the encoded data. In order to use the resulting sparse matrices, another processing tool unit must be designed. An example is a classifier described in Section 4.3. An alternative solution, which is not presented in this work, could be to train a CNN directly performing a specific task (e.g., classification) with enforced sparsity inside. In this case, however, the sparse information would not preserve the whole information about the input, but only this part which is required to accomplish a given goal.

## 3. Method

### 3.1. Method Overview

As proposed by LeCun [2], CNNs are feed-forward neural networks that typically consist of the following.

- Convolutional layers, which consist of multiple groups of matrix convolution filters. Input channels are convolved with corresponding filters from a group, and the sum of convolutions is a single output channel. The number of output channels is equal to the number of filter groups.
- Pooling layers, which divide the input image into a grid and reduce each cell to a single pixel. A commonly used pooling option is max-pooling, which takes the maximum value of each cell.
- Processing units such as activation functions or regularization techniques. The latter usually operates only on the gradient values used in the learning process, which can be used to implement additional components of the cost function.

In this paper, two applications of CNNs are considered. The most important architecture proposed in this study is a CNN-based autoencoder. Its primary goal is to minimize the difference between the input data and the output obtained for any input from the considered data set. The difference is calculated as the Euclidean distance between vectors of pixels. This implies that the resolutions of both input and output are expected to be the same. Thus, in our approach, no pooling layers are used and each convolutional layer is complemented by appropriate padding. As the convolution of $m_w \times m_h$ matrix with $f_w \times f_h$ filter yields $(m_w - f_w + 1) \times (m_h - f_h + 1)$ as a result, $(f_w - 1)/2$ zero-padding is added to the sides of the matrix, and $(f_h - 1)/2$ to the top and bottom. This is possible when both filter dimensions ($f_w$ and $f_h$) are odd. This property is illustrated in Figure 1.

Without setting additional requirements, it would be easy to construct a perfect CNN-based image autoencoder. It would be sufficient that each layer generated an output equal to the input. This could be achieved with a convolution filter that has 1 in a single matrix element and 0 everywhere else. In order to avoid a meaningless result like that, we force one selected hidden layer to consist only of sparse matrices. The sparsity is guaranteed in the following way. For each non-zero element $(i, j)$ of

the output matrix, all other elements in $s \times s$ square centered in $(i, j)$ are reduced to zeros. This step of data processing is further referred to as local-maximum filtering (LMF) (Figure 2) and its details are described in Section 3.3.

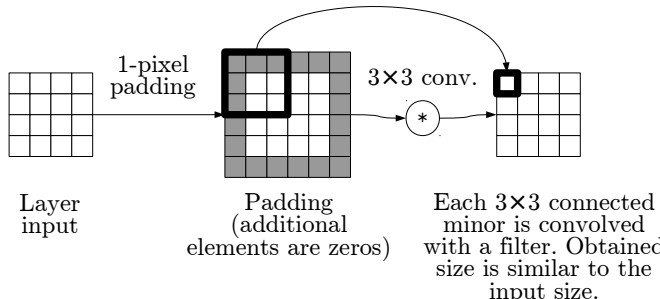

**Figure 1.** The operation illustrated in this figure is a superposition of $1 \times 1$ zero-padding and matrix convolution with $3 \times 3$ filters. As the padding size matches the filter size properly, the output matrix has the same size as the input matrix.

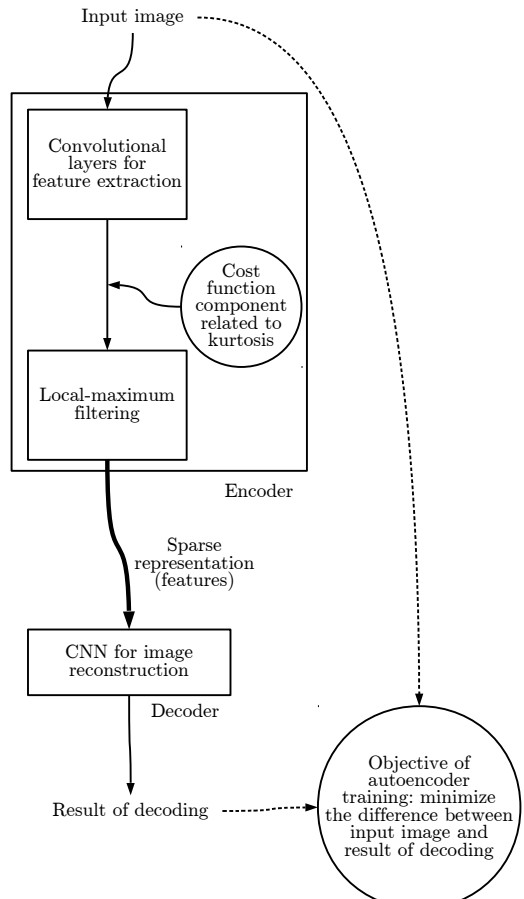

**Figure 2.** The full architecture of the autoencoder consists of two major parts: encoder and decoder. The encoder includes convolutional layers that can either be adjusted to the data set in the learning process or use some fixed weights. The result is further processed with local-maximum filtering (Section 3.3). In the case of adjustable convolutional layers in the encoder, the cost function related to the encoder's CNN output is modified in order to reduce the output kurtosis, as described in Section 3.2. The encoder output is fed into the decoder which consists of convolutional layers that participate in the learning process. The learning objective is to reproduce the original input image, while minimizing the reconstruction error, which is measured in terms of the Euclidean distance.

### 3.2. Leptokurtic Feature Maps

In order to obtain satisfactory results of local-maximum filtering in the selected hidden layerduring CNN training, the learning objective is enhanced with a kurtosis-based adjustment. It is based on the following observation; splitting the convolutional layer outputs into small subsets of highly activated points and low activation of the other elements may be equated to leptokurtic distribution of the outputs. Leptokurtic distribution (related to high kurtosis) means that all the elements are concentrated closer to the mean value than in the case of normal distribution. Forcing the leptokurtic distribution may be considered as a process equivalent to kurtosis maximization. The kurtosis function is continuous and differentiable almost everywhere, which provides the ability to apply gradient-based learning. Consequently, it can constitute an additional component of the cost function.

The kurtosis [30] of a vector $X = (X_1, X_2, \ldots, X_n)$ (this notation is valid for both random variables and fixed numbers) is defined as

$$\text{Kurt } X = \frac{\mu_4(X)}{\sigma(X)^4} - 3, \tag{1}$$

where $\mu_4(X)$ is the fourth central moment and $\sigma$ is standard derivation. In order to perform the gradient learning, we need to calculate the actual gradient. As the formulas are symmetric in terms of the elements of $X$, the only expression we need is

$$\begin{aligned} \frac{\partial(\text{Kurt } X)}{\partial X_i} &= \frac{4\left(X_i^3 - \text{E}((X - \text{E}X)^3)\right)\text{Var}(X)}{n\text{ Var}(X)^3} + \\ &\quad - \frac{4\text{ E}((X - \text{E}X)^4) \cdot X_i}{n\text{ Var}(X)^3}. \end{aligned} \tag{2}$$

Formula (2) has one important disadvantage when used for gradient learning. As kurtosis (1) is indifferent to the magnitude of the inputs, the differential decreases as the magnitude of the inputs grows. As a result, big values would be modified more slowly by the learning process. In order to reverse this effect and obtain a change that is proportional to the current value of convolutional layer outputs (and to the corresponding weights—as in the case of CNNs these terms are proportional), the differential (2) is multiplied by $\text{Var}(X)$. This means using exponent 2 instead of 3 in the denominators of expression (2).

### 3.3. Local-Maximum Filtering

The additional gradient component, which makes the selected part of the CNN yield leptokurtic outputs, does not guarantee the desired properties of the sparse output. In order to achieve literal sparsity, we need to make sure that some matrix elements are replaced with zeros. This could be easily achieved by thresholding—a process similar to that described in [31]. In order to limit the number of the remaining outputs, the threshold level could be defined as a quantile of the output of either the whole layer or a single resulting matrix. In this work, however, instead of using the global statistics of the CNN layer output, we propose a method that focuses on local properties.

The proposed approach, which generates only one non-zeroed element in each $s \times s$ matrix minor, has two major advantages. First, this operation is easy to implement for parallel computations, which is important, as the present CNN solution was implemented using GPU, supported by the Caffe framework [32]. Each element is considered separately, and is zeroed if it is not strictly the greatest element in the surrounding square. Another advantage of the proposed way of forcing the sparse representation is related to the interpretation of visual features. Typically, the input image has a continuous content, which means that the same visual feature is likely to be detected in multiple neighboring locations. Let us consider a horizontal edge visible in the image as an example. In the case

of a long horizontal line in the image, the same local feature is obviously present in all the points of this line. If the sparse representation was related only to the number of activated pixels, there is a strong probability that we would obtain a subset of pixels forming a single connected component around the most visible feature (or, as in the example, along the line). LMF provides a direct solution to this problem. This situation is illustrated in Table 1.

**Table 1.** Local-maximum filtering (LMF) is a method that generates a sparse output, but is more practical than the standard thresholding. The points are designed to reflect the selected visual features of the image, and the local-maximum filtering makes it possible for each point to reflect a different occurrence of a feature. Thus, it is necessary to employ a mechanism preventing non-zeroed points from being located too close to each other. The strongest activated points are chosen in a greedy way, with only one point allowed in each $s \times s$ square. The presented illustration shows the result for $s = 3$.

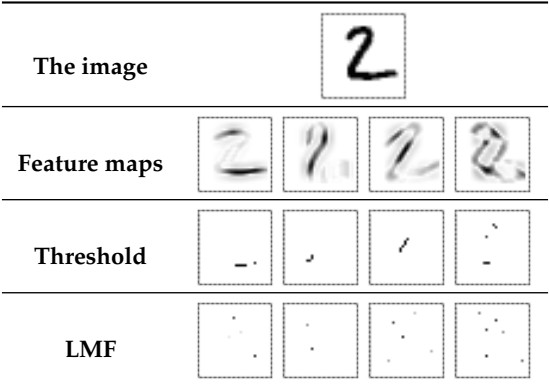

### 3.4. Additional Thresholding

Local-maximum filtering, which was described in the previous section, generates an output that can be regarded as sparse. In the case of the MNIST data set [7], where input data consists of $28 \times 28$ images, local-maximum filtering with radius $s = 3$ ensures that at most 49 elements of each $28 \times 28$ matrix remain non-zeroed. This means that either for the original data set or any larger input images, at most 6.25% elements of the output data have values other than zeros. It may be expected, however, that many of the 49 elements have insignificant, near-zero values anyways. Local-maximum filtering yields such an element in each isolated region of the image plane, even if the corresponding visual feature is not present in that region.

It is difficult to suggest any general purpose threshold for the selection of the significant points, as it may depend on the weights of the convolutional layer and on the context of the considered image. In some of the experiments that involve the original MNIST data set, where each image presents exactly one object, we manually limit the number of points in each matrix that are used to encode that object. After local-maximum filtering, which prevents the points from being located too close to each other, all the points except the $k$ highest values are replaced with zeros. For $k = 5$ and $k = 3$, it yields 0.64% and 0.38% non-zero values, respectively. This is equivalent to image thresholding, with the threshold value dependent on the appropriate quantile of the values from the processed matrix. This highly sparse representation can be used to experimentally determine how much information is actually preserved in the small number of points.

### 3.5. Properties

The sparse output obtained from the selected hidden CNN layer can be considered as a form of image encoding, as it is supposed to be used by subsequent layers to reconstruct the input image. Thus, a neural network architecture that meets the presented assumptions can be considered as a general purpose tool for sparse image encoding. The encoding is based on local visual features of the image, which may be easily explained as follows. One of the commonly known properties of

CNNs is the invariance to translation of the visual features of objects on the input image plane [2]. The translation of the object automatically results in a similar translation of its representation generated by the convolutional layer. This is essentially true for a single matrix convolution and remains relevant for sums and superpositions thereof. The outputs of the convolutional layers are known as feature maps, as CNNs take a biological inspiration from the visual cortex [2,33]. An important aspect is the size of the feature or object visible in an image under examination. Calculations performed in order to obtain each element of the feature map involve data from a specific range of the input image, known as the visual field. In the case of the initial convolutional layers, the visual field is significantly smaller than the image itself—for the first layer, it is simply equal to the filter size. If the visual fields of the layer with a sparse output contained the whole input image, whole objects could be encoded as single pixels. However, this would be equivalent to an image classification task, without the analysis of particular elements of the recognized object. In order to split the original image into more basic features, we use visual fields that are smaller than the image itself. In one of the examples, presented in the following section, we use $14 \times 14$ visual fields selected from $28 \times 28$ input images. The sizes of visual fields of a neural network are easy to estimate, particularly if the network consists of convolutional layers only, an example is shown in Figure 3.

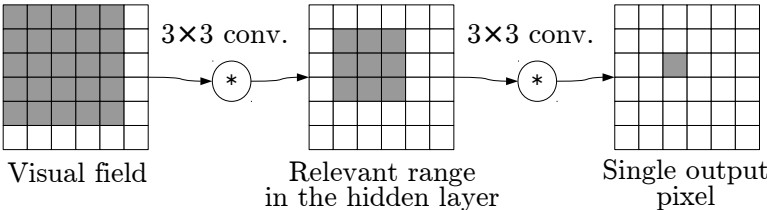

**Figure 3.** Each output pixel depends on multiple elements from the previous layers. The scope of the related pixel from the previous convolutional layer matches the size of the convolutional filter applied. By tracking the dependencies back to the input data matrix, we can determine the size of the visual (receptive) field. The convolutions involve one-pixel padding from each side, so the image size does not change. A single output pixel is calculated on the basis of $3 \times 3$ minor of the hidden layer, and the size of the visual field is $5 \times 5$.

## 4. Experiments

### 4.1. Feature Identification

The experiments described in this section were performed on the original MNIST data set [7], which offers the advantages of a large number of images, a resolution that makes the computational cost considerably low ($28 \times 28$ pixels), and a simple semantic interpretation of the results, as the samples contain handwritten digits.

Three experiments were performed in three set-ups that implemented the idea presented in Figure 2. According to the original partition of the MNIST data set, the autoencoder models were trained with the 60,000 training samples, while the separate 10,000 samples were used for the evaluation. The models were different in terms of the visual features used to encode the image. The architecture of the decoder part, described in Table 2, was common for all the models. None of the models used any form of pooling, and the coexistence of filters and paddings made the matrix size remain unchanged throughout the layers. The presented models applied typical techniques associated with CNNs, such as the dropout method [34] and PReLU activation functions [35]. The encoders were designed as follows.

- MF4: Four manually designed features were used. The filters were fixed and no learning was performed on this encoder. The features were related to vertical, horizontal, and diagonal lines (in both diagonal directions). The contents of the proposed feature-detecting convolution filters are presented in Table 3. The filters applied in this experiment are of a very generic character,

so no advantage may be drawn from using specific filters that fit the data set. Absolute values of the matrix convolution results are used, which is followed by local-maximum filtering with radius 3, as described in Section 3.3.

- AF4: Four automatically computed features. The architecture of the encoder (or feature extractor) is described in Table 4. The size of the visual field (Section 3.5) for this architecture is $14 \times 14$. The special activation function used in the last layer of this architecture is denoted as ENCODE. The outputs of Enc5 are tuples of four sparse matrices, considered as extracted features of the input image.

- AF5: Five automatically computed features. This experiment is largely similar to AF4, but five matrices are generated as the output of Enc5 layers (instead of four in AF4).

The encoder and decoder could be considered as separate utilities, but combining them into one neural network model made it possible to actually train the feature detectors in AF4 and AF5 experiments. The training was aimed at minimizing the total square error of the autoencoder.

The MF4 features are the most natural approach, as the features were designed manually in order to approximate any pattern that consists of thin lines. The four basic directions, shown in Table 3, fit the structure of a filter matrix precisely. Any change to this approach, such as a set of 3 or 5 segment-based features, would require an arbitrary choice of a direction and involve a specific approximation when described as convolutional filters.

As MF4—a solution with 4 kinds of features—was selected for its simplicity, the most direct comparison based on the automatic features identification involves 4 features as well, which is demonstrated by the AF4 set-up. However, automatic detection of features does not directly indicate any specific number of features as correct. The design of 5 equally important features for MNIST is unintuitive, but the potential gain can be easily researched for using the automatically trained encoder. The AF5 set-up was introduced for this purpose. The number of features can be expanded arbitrarily further, but as the number of features would grow, they would be increasingly difficult to visually distinguish. For the purpose of visual presentation of the results, we focus on a maximum number of 5 features. However, if the data set was more complex than MNIST or involved color images, it may be crucial to introduce more features.

The specifics of automatic encoder training process that are described in Table 4 were proposed as a compromise. This architecture is complex enough to identify potentially useful image features while avoiding the possible disadvantages of overly complex models, such as high resources usage and duplicated filters. The number of layers and the filter sizes were defined by the requirements on the visual fields, while the number of filters in each layer was selected by trial and error. The results were roughly convergent around the chosen preferred values. The possible changes obviously include permutations of feature detectors in the encoder output. The full training time was long enough to make the detailed parameter tuning remarkably difficult, but we believe that the presented models are sufficient to demonstrate the properties of the proposed methods.

The complete encoder architecture from Table 4 involves 28,570 adjustable parameters for AF4 and 29,570 for AF5. The slight difference is related to the last convolutional layer in the sequence. Remaining in a similar order of magnitude, the total number of decoder weights was 114,450 for AF4/MF4 and 115,200 for AF5, due to the additional filter group in the first convolutional layer. While the training process was relatively complex, we believe that the final model can be described as lightweight.

It must be emphasized that getting the optimal autoencoders available to this method would require much more detailed fine-tuning and repeated experiments. However, the presented demonstration of the method does not require putting this kind of endless effort to the optimization. We have defined three different set-ups, which are going to be useful for the analysis, and we use fixed training conditions for all of them, so we can adequately compare them with one another.

The autoencoder error was calculated as the difference between the input and the expected output. Data from the MNIST data set [7] could be considered as a set of 8-bit grayscale images with brightness

levels varying from 0 to 255. However, the presented results refer to normalized values from the [0, 1] range. This applies to the average errors from Table 5. The error for a single sample is a half of the sum of quadratic errors for all the pixels. The table presents average errors for a certain set of samples—both for the whole data set and for all the digits considered separately. The autoencoder itself, in accordance with the previously described architecture, did not use any information on object classes while training.

**Table 2.** All the experiments (namely, `MF4`, `AF4`, and `AF5`) related to the general architecture of the autoencoder, which is shown in the Figure 2, use the same layout of the decoder part. This table includes a detailed layout of the convolutional layers and the activation functions used in the decoder, such as PReLU [35]. The rows of the presented table describe consequent layers of the decoder CNN, denoted as `Dec1`–`Dec4`.

| # | Outs | Filter | Pad | Dropout | Activation |
|---|---|---|---|---|---|
| `Dec1` | 30 | $5 \times 5$ | $2 \times 2$ | - | PReLU |
| `Dec2` | 30 | $5 \times 5$ | $2 \times 2$ | - | PReLU |
| `Dec3` | 30 | $5 \times 5$ | $2 \times 2$ | 0.5 | PReLU |
| `Dec4` | 30 | $5 \times 5$ | $2 \times 2$ | 0.5 | PReLU |

**Table 3.** One of the autoencoder-related experiments, labeled as `MF4`, uses fixed encoder filters (the encoder part of the overall architecture is described in Figure 2). This table presents the predefined values of $7 \times 7$ convolutional filters, visualized as bitmaps.

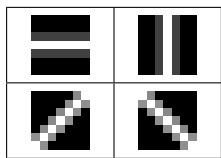

**Table 4.** Automatic feature extraction experiments, labeled as `AF4` and `AF5`, use adjustable encoders (Figure 2) with multiple convolutional layers. The layout of the layers and the corresponding activation functions (including PReLU [35]) are presented. The ENCODE activation function is a short term for a sequence of operations: the PReLU activation function, the absolute value, the layer that modifies gradients with relation to kurtosis (Section 3.2), and local-maximum filtering with radius $s = 3$.

| # | Outs | Filter | Pad | Dropout | Activation |
|---|---|---|---|---|---|
| `Enc1` | 50 | $3 \times 3$ | $1 \times 1$ | - | PReLU |
| `Enc2` | 30 | $3 \times 3$ | $1 \times 1$ | - | PReLU |
| `Enc3` | 30 | $3 \times 3$ | $1 \times 1$ | - | PReLU |
| `Enc4` | 30 | $3 \times 3$ | $1 \times 1$ | - | PReLU |
| `Enc5` | 4 or 5 | $5 \times 5$ | $2 \times 2$ | - | ENCODE |

**Table 5.** The autoencoder errors, measured in terms of the Euclidean loss function, were calculated for all the proposed network architectures. In addition to the general error on the test set from the MNIST data set [7], specific values were calculated for each class separately. Thus, it was possible to evaluate how well the selected visual features described each of the digits.

| Class | MF4 | AF4 | AF5 |
|-------|--------|--------|--------|
| **0** | 14.624 | 9.852 | 8.820 |
| **1** | 7.248 | 5.056 | 4.500 |
| **2** | 12.210 | 9.308 | 9.386 |
| **3** | 11.726 | 8.348 | 7.966 |
| **4** | 10.262 | 8.136 | 7.760 |
| **5** | 12.168 | 8.876 | 8.114 |
| **6** | 11.802 | 9.372 | 8.780 |
| **7** | 9.986 | 6.592 | 6.266 |
| **8** | 12.656 | 10.458 | 10.198 |
| **9** | 10.260 | 7.682 | 6.732 |
| **All** | 11.220 | 8.304 | 7.792 |

The results presented in Table 5 prove a relative success of all the experiments. It is worth noting that the maximum quadratic error between $28 \times 28$ matrices is 784, and the expected quadratic difference between matrices of uniform random $[0, 1]$ elements is 130.67. The errors obtained from the experiments presented are lower by a whole order of magnitude (per-subset average errors are more than 11 times smaller than the mentioned estimation). The only limitation, which leads to the presumption that error of zero is impossible for the MNIST data set, is based on the sparse encoding that needs to be used as an intermediate sample representation. Due to the specific properties of this sparse representation, there is no other comparison. The training of the decoders was performed in a unified way for all the set-ups, so the results from Table 5 reflect the usefulness of features selected by the encoders. Therefore, in absence of more general ground truth, MF4 results can be considered as reference values for evaluation of AF4- and AF5-based features.

The first conclusion is that the features specific to the data set performed better than the generic manual suggestion—the MF4 experiment resulted in the highest autoencoder errors. The difference between the results of the automatic variants with 4 and 5 features appears to be slight when compared to MF4. It may be also concluded that using a higher number of features makes the encoding more precise, i.e., it enables preserving more information about the exact contents of the original image. Surprisingly, the results for digit 2 are slightly better in the case of AF4 than in AF5, which is an exception to the mentioned rule.

The differences between classes can be explained by the geometric properties of the digits. Digit 0, which is round, generated a particularly high error in MF4, as lines of fixed directions made it difficult to recognize round shapes. The error for 0 in MF4 was even higher than for 8, which contains crossing diagonal line segments in the center—the direction of these segments apparently fits the designed filters. Remarkably, the lowest errors for MF4 were obtained for digits that literally consist of straight segments, namely, 1 and 7. While digit 9 was the third best, 4 was the close fourth, which fits the pattern, as 4 consists of long segments and 9 has a small circular head and a straight, long tail.

The comparison of AF4 and AF5 error rates provided a number of other important observations. In both experiments, 8 was the worst case, which can be justified by the most visually complex shape—a single line that crosses itself and forms two circles is especially difficult to describe with features obtained as the result of convolutional filters. The other digits with significantly high error rates were 0, 2, and 6. For MF4, the digits that contained circles (0 and 6) produced high error rates, while for MF5, the second worst case was 2. It suggests that MF5 was able to handle the features

characterized with small circle shapes better than MF4, partly at the cost of segments specific to digit 2. The difference between errors obtained in MF4 and MF5 is the highest for 0, 9, 5, and 6; remarkably, three of these digits have shapes containing circles.

As was the case in MF4, and also in the other experiments, the lowest error rates were generated for 1 and 7. The property of these digits, which can be summarized as *having a simple shape*, seems to be pretty universal, as confirmed by the results obtained for AF4 and AF5.

### 4.2. Feature Reduction

The experiments presented in the previous section involve local-maximum filtering, which ensures that at most 6.25% of matrix elements are non-zeros. In this section, however, the results related to even higher levels of sparsity are considered. The number of zeroed elements in the encoding is increased, but exactly the same decoders, trained in Section 4.1, are used to generate the results presented below.

Figures 4–6 include the results of sparse matrices decoding for MF4, AF4 and AF5 experiments respectively. Each table includes the following.

- The original encoding errors (for comparison).
- The result achieved with each matrix being greedily reduced to 5 highest values, and all the other elements being replaced by zeros. The description of there results consists of an absolute encoding error and a relative increase (compared to the first column).
- The results of an experiment similar to the previous point, but with 3 points instead of 5.

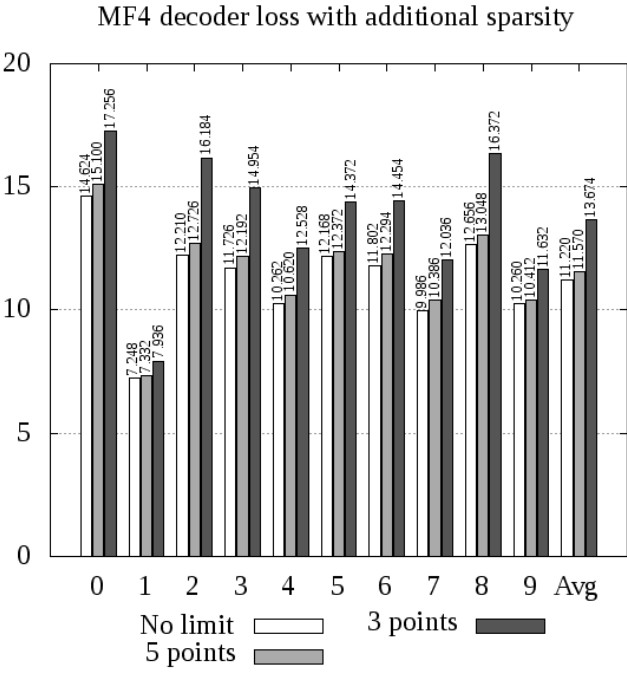

**Figure 4.** The pretrained decoder from the MF4 model can be used either with the original data without a specific limit of non-zero elements in the encoding, or with modified encodings, where each matrix contains up to 3 or 5 non-zero elements. The plot presents decoding errors for images showing individual digits and for the whole test set.

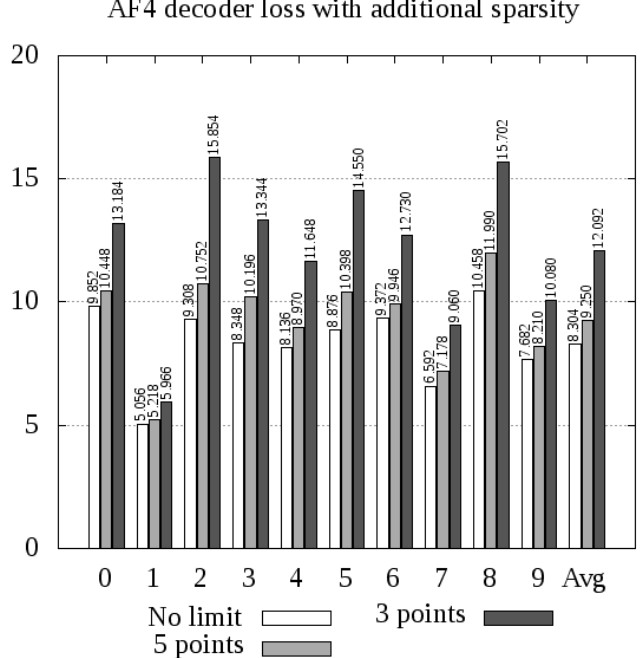

**Figure 5.** The pretrained decoder from AF4 used for decoding of both the original and the highly sparse data, as in Figure 4.

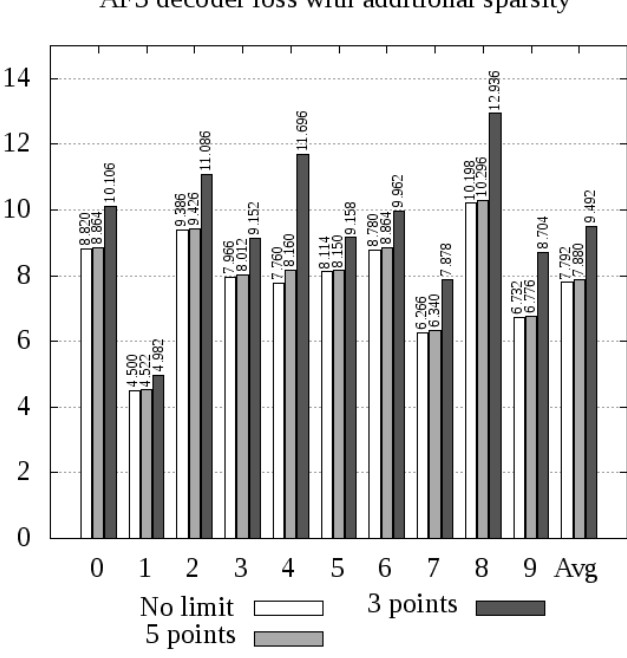

**Figure 6.** The pretrained decoder from AF5 used for decoding of both the original and the highly sparse data, as in Figure 4.

As we can conclude from Figures 4–6, experiment AF4 seems to be most sensitive to additional thresholding, which is particularly evident in the case of encoding digits 2, 3, and 5. However, the other experiments, namely, MF4 and AF5, behave in quite a similar way, giving slightly above 20% greater average loss when 3 points per matrix are used, and only a few percent in the case of 5 points.

The most remarkable phenomenon related to experiment AF5 is the sensitivity of digit 4 to sparse autoencoding. The autoencoder error increased by 5% for 5 points and above 50% for 3 points. This leads to the conclusion that digit 4 consists of a greater number of visual feature occurrences than any other digit, and omitting some of these features generates a significant error.

The most important conclusion is that further sparsity enforcement is generally acceptable, unless the features are too specific (AF4 case) and the reduction is too great (3 points case). With 5 points per matrix, both MF4 (error increase: up to 4%, 3% in average) and AF5 (error increase: up to 5%, only 1% in average) models yielded acceptable results. This means that the whole $28 \times 28$ digit can be compressed into 20 points (in the case of MF4) or 25 points (AF5), with encoding errors presented in Figures 4 and 6.

### 4.3. Classification

In order to determine how much information was preserved in the encoding, we attempted to decode the original image, as described in the previous sections. However, it is not the only possible approach. It is debatable whether the Euclidean distance between the autoencoder output and the original image may serve as a reliable tool for measuring the loss of significant information in the encoding process. However, regardless of the Euclidean distance value, the encoding can be considered as useful if it is sufficient to determine the originally encoded digit. This property can be tested in the image classification task using pre-generated encodings. Another reason for performing this experiment is the possibility to discuss the relation of our results to the numerous classification results from the literature, where a similar task was performed on the same data set.

The sparse representations obtained from the encoder (according to the description shown in Figure 2) can not only be used to decode the original digit, but also directly in the image classification task. All the experiments (MF4, AF4, and AF5) were performed on the basis of a CNN classifier architecture proposed by the authors of this study. The classifier consisted of 6 convolutional layers and two hidden fully connected layers. The last convolutional layer and the hidden fully connected ones were trained using the dropout method [34]. Such an approach was decided, as it should provide adequately complex classifier model to achieve fine results without defining a very deep neural network which would require specific approach to the problem of a vanishing gradient. A model with 30 convolutional filters in each layer and 500 neurons in the hidden fully connected layer was selected as a point where no further extension improved the result significantly. The presented values indicate that the trained classifier models consisted of less than 50,000 convolutional parameters and approximately 12 millions of weights of the fully-connected layers. It must be emphasized that finding the optimal classifier model was not the key objective of this paper. The selected classifier configuration is possibly similar for all the encodings, and the results are well adjusted to the task of comparison between the setups. Further effort to optimize such a classifier remains possible, but this issue alone definitely exceeds the scope of this paper.

For the classification tests, the data set was divided into a testing set (10,000 samples) and a training set (60,000 samples), as proposed in the original MNIST [7] database. Each classifier was tested with a representation obtained by a specific autoencoder. This architecture made it possible to perform additional experiments. Instead of a raw encoder output, where up to 49 pixels from each matrix could have positive values, manually thresholded matrices were used in order to eliminate near-zero values. The data prepared in this way are used in the same tasks as described in the previous sections. It must be emphasized that the same classifier models were used for both the original encodings and the thresholded versions. All the results are presented in Table 6.

**Table 6.** Encodings from Section 4.2 were tested in the image classification task. Each encoding was used both in the original form, obtained as a result of local-maximum filtering, and in the reduced form that guarantees additional sparsity (Section 3.4). For each encoding (MF4, AF4, and AF5) a separate classifier was trained. The results for the additionally thresholded data were generated with the same neural networks that were trained for the original encodings.

| Model | Accuracy |
|---|---|
| MF4 (3 points) | 93.84% |
| MF4 (5 points) | 95.59% |
| MF4 | 95.64% |
| AF4 (3 points) | 96.86% |
| AF4 (5 points) | 98.33% |
| AF4 | **98.67**% |
| AF5 (3 points) | **97.88**% |
| AF5 (5 points) | **98.40**% |
| AF5 | 98.40% |

As we can deduce from Table 6, the accuracy of the classifier seems to reflect the autoencoder error from the previous tables. Thus, MF4 results are clearly the worst—the general features are not nearly as useful as the automatically calculated ones that were used in experiments AF4 and AF5. The only surprise is that the AF4 classifier on the full encoder results was the best from the whole table (98.67%)—the difference is slight, but the classifier related to AF5 made more mistakes. However, when the reduced representations are considered, the sensitivity of the representations encoded in AF4 to the additional thresholding is clearly visible, as was the case with the autoencoder. AF5 representations, when reduced to five points per matrix, resulted in as good results as in the case of the original classifier objective, providing an accuracy of 98.40%. Surprisingly, the representations reduced to 3 points per matrix, despite generating over 20% higher autoencoder error, can still be regarded as acceptable for practical applications, as even with so drastically reduced information the classifier is able to recognize the digit correctly in 97.88% of the cases.

As the results from Table 6 are denoted as classification accuracies that can be easily compared to each other, we can seek comparison with other MNIST classifiers from the literature as well. However, it must be emphasized that in this paper we treat the image classification just as an analytic tool, and not as the key objective of this paper. Presumably, using the raw MNIST images to train a classifier, without the added difficulty of sparse representation, could only improve the achieved accuracy. The general problem of MNIST classification can be solved with accuracy as great as 99.79% [36]. We do not pursue to beat this result. For broader perspective, we can discuss the relation of our results to the other state-of-art MNIST classifiers that somehow involve sparse representations. Due to the varying objectives and circumstances, such comparisons require analysis that exceeds the straightforward competition for the best accuracy.

The results from Table 6 are clearly better than the classification results obtained with the classical approach to sparse representations and dictionary learning. This includes particularly the convolutional sparse coding for classification (CSCC) method presented in [23], which achieved an accuracy of 84.5% on the MNIST data set, outperforming many previous approaches to sparse representations and dictionary learning. It must be emphasized, however, that the problem statement of that paper was not the same as ours. Dictionary-based methods are more computationally complex. Moreover, in [23], only the training subsets of 300 images were used. Thus, while that work may be regarded as an interesting reference for the present study, a direct comparison would be inappropriate.

Another remarkable work on sparse representation was based on the idea of maximizing the margin between classes in the sparse representation-based classification (SRC) task [25]. The sparse

representations related to this model were strictly related to the classification task. In contrast to that approach, the method presented in this paper does not use any information on object labels when training the encoder. On the other hand, no convolutional neural networks were used in [25], and some solutions used in that paper might be outdated. The best classifier presented there reached a 98.13% accuracy rate. This result is lower than AF5 with 5-point-based reduction, which is already very sparse.

The CNN-based architecture ensures that the image features are detected is a translation-invariant manner; translation of a feature would entail translated coding. A similar concept was applied in [22], which proposed another approach to CNN-related sparse coding. The results of MNIST classification were generated for both the unsupervised and the supervised approach to sparse coding, with 97.2% and 98.9% accuracy rates, respectively. It must be emphasized, however, that the method shown in the present paper should be considered as unsupervised, as the autoencoder does not use information on the object labels. The size of the input data is not fixed—the method works for any input data, irrespective of the number of rows and columns. Thus, we cannot speak of a class that an input belongs to and some valid input images can contain multiple digits, which makes it impossible to assign them to a single label.

The results of MNIST classification that are somehow related to the idea of using sparse coding in the hidden layers in image processing tasks are also known from the works on spiking neural networks. A solution which involved weight and threshold balancing [31] performed reasonably well, resulting in a 99.14% accuracy rate in a method that combined spiking neural networks and CNNs. However, the method proposed in [31] was very complicated and the image representations that it produced were not as sparse as those presented in this paper. Similar remarks hold with respect to the work in [28] (non-CNN spiking network with LIF neurons) and the work in [31] (bio-inspired spiking CNNs with sparse coding), which achieved the accuracy of 98.37% and 97.5%, respectively. The latter approach is particularly interesting, as it was coupled with a visual analysis of features recognized by the neural network. The accuracy rates achieved were slightly lower than these obtained in this paper. However, the results in [31] cannot be directly compared with these achieved in this study because of differences in the architectures proposed. Moreover, the work in [31] involved an additional learning objective—the classifier was designed and trained to handle noisy input.

### 4.4. Larger Images

All our previous experiments were related to the original MNIST data set [7], where each sample was a 28 × 28 image displaying a single centered digit. The autoencoder was designed to encode each digit in a way that enabled as accurate a reconstruction of that digit as possible. As the solution is based on CNNs (both encoder and decoder, as it is shown in Figure 2), the whole mechanism is translation invariant—-a translated digit would simply yield a translated sparse representation. What is more, as no pooling layers are used, the model can be successfully applied to images of any size. Both matrix convolutions and element-wise operations will still be possible to be computed.

The modified data set with larger images was prepared to illustrate this property, as shown in Figure 7. The digits were placed on 80 × 80 plane in a greedy way, as long as placing another non-intersecting 28 × 28 square was possible. The test set consisted of 2000 images: 68 with a single digit, 119 with two digits, 888 with three digits, and 925 with four digits each.

The features described in the proposed sparse representation are deliberately smaller than the whole digits, so our model should not be considered as digit classifier, in particular for larger, more complex images. Nevertheless, digits should be reconstructed equally well regardless of position and context. The experiment introduced in this section is intended to demonstrate this property.

Table 7 shows the average per-digit autoencoder errors for the extended data set. In the case of images with multiple digits, the error was divided by their count. The division into separate classes was impossible, as a single large image was likely to contain multiple digits from different classes. The overall conclusion is that the MF4 model is quite sensitive to the behavior of image boundaries and, while useful, produces almost 10% greater errors in this atypical application. The models with

automatically calculated features—AF4 and AF5—provided a very slight error increase when compared to the original task. This confirms the universal nature of the presented autoencoders. As expected, translational invariance makes it possible to describe the translated objects as easily as the original inputs. The application of extended input sizes does not create any technical difficulties either.

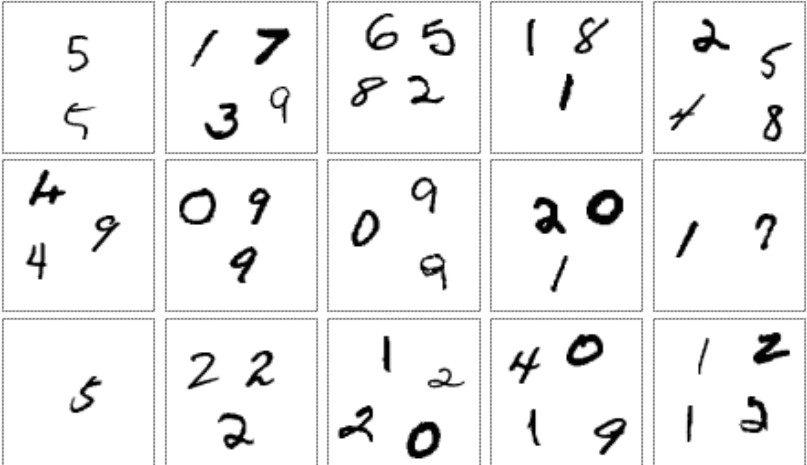

**Figure 7.** In order to demonstrate that the proposed autoencoder architecture is size-independent, 80 × 80 images containing multiple digits from the MNIST data set [7] were generated. The extended images contain up to four objects.

**Table 7.** The autoencoders trained in Section 4.1 were used with larger images, as shown in Figure 7. An average per-object error was calculated and compared to the original results, related to the objects with a single centered object. The information on the relative error increase is included in the table as well.

| Model | Original | Large | Increase |
|-------|----------|-------|----------|
| MF4 | 11.220 | 12.220 | 8.91% |
| AF4 | 8.304 | 8.342 | 0.46% |
| AF5 | 7.792 | 7.902 | 1.41% |

## 5. Analysis

The autoencoder architectures presented in Section 4.1 can be further analyzed in terms of errors and semantic understanding of related visual features. The results from Sections 4.1 and 4.2 can be used to compare the overall quality of selected solutions and analyze the dependency between the autoencoder errors and particular image classes. This could be related to the level of adjustment of the set of selected features to the data set, for example, manually selected directions of lines are particularly irrelevant in the case of digits 0 and 5. Another important aspect is the inner complexity of the digits. Digit 1, which usually consists of one or two segments, is especially easy to model. The exact directions of the segments, however, do not fit any of the manually selected filters. Thus, only an automatically computed feature extractor was able to take full advantage of the simplicity of the shape of this digit, producing a remarkably low error rate for this class. The errors for the other nine classes differ only slightly in the case of the automatically chosen features, which means that this solution actually reflects the properties of the data set. Manually selected features were not digit-specific, which resulted in generally higher error rates and greater variance of errors among different digits in MF4 experiment.

In the case of the manually designed features, the convolution filters were created arbitrarily, as illustrated in Table 3. However, another way of visual presentation of the features can be achieved by using the decoder part of the autoencoder architecture (see Figure 2). The results of decoding

single points related to each of the manually chosen features are presented in Table 8—the relation to the filters shown in Table 3 is apparent. This technique of visualization can also be applied to the models with the automatically constructed features. The results of this approach are reported in Tables 9 and 10—apparently, this time the features are implicit and difficult to categorize semantically.

As the shape of each visual feature is complex, and possibly context-dependent, we need an analysis that goes beyond simple visualization. Tables 11–13 provide information on the intensity of each feature in the data set, including both averaged results and those obtained for separate subsets consisting of different digits. The intensity is calculated as a sum of the elements of the respective encoding matrix (output of the encoder, as illustrated in Figure 2). It must be emphasized that because of different weights in neural network models related to each decoder/classifier, it is possible to compare only the values within the same table. However, the results presented still enable a thorough analysis that otherwise would be difficult to perform.

The MF4 results (see Table 9) are especially easy to understand, as digits are rather taller than wide, vertical lines (feature #2) are the most visible among the data set. Remarkably, digit 1 contains almost no other features. The least intense feature is related to the backslash segments (#4), which occur mostly as a part of arc (digits 8, 3, and 0). As it was expected, digit 8 is especially rich in all kinds of features. However, because of the typically skewed writing style, even in this particular case backslash lines are less intense than segments in the other directions.

The features selected in the AF4 model are particularly interesting. It is relatively clear that no feature is dedicated solely to backslash lines, which is demonstrated in Table 10. Instead, we get feature #1 that seems to address the right side of a small arc. This is reflected in Table 11—digits 8, 6, and 3 exhibited particularly intense occurrences of this feature. While features #2 and #3 seem to reflect vertical and horizontal lines, respectively (digit 1 is strongly correlated with feature #2), the relation is not nearly as straightforward as it was in MF4. Feature #4, related to slash lines, seems to reflect a part of digit 2 especially well.

The results of the experiment AF5, which yielded the best autoencoder (Section 4.1) and classification accuracy for highly sparse data (Section 4.3), are less intuitive. The idea of horizontal lines is divided between features #2 and #4. Features #1 and #5 seem to handle both slash/backslash lines and sections of small arcs as well—both these features are important for encoding digit 8. Feature #3 clearly describes some cases of curves that are oriented vertically (digits 2, 8, and 6), but it does not involve straight segments. Digit 1 is described mostly with feature #4. Visual features detected by AF5 model are mostly implicit and difficult to describe semantically.

**Table 8.** MF4 decoder results for synthetic encoding, where only one point is activated. This is intended to show the shape of the visual features that are described by each matrix.

| | Code | | | | Output |
|---|---|---|---|---|---|
| #1 | ▪ | | | | — |
| #2 | | ▪ | | | / |
| #3 | | | ▪ | | / |
| #4 | | | | ▪ | \ |

**Table 9.** For each visual feature MF4, the sum of occurrence intensities (values in the encoding) was calculated. The result was considered separately for each digit, as different digits consisted of different visual features. It must be emphasized that the results in this table are relative and, while comparing them to each other is noteworthy, they should not be compared with the results from the other tables.

|      | #1   | #2   | #3   | #4   |
|------|------|------|------|------|
| **0**    | 1.59 | 1.82 | 1.88 | 1.86 |
| **1**    | 0.52 | 1.53 | 0.88 | 0.87 |
| **2**    | 2.16 | 1.69 | 1.89 | 1.79 |
| **3**    | 2.28 | 1.69 | 1.94 | 1.90 |
| **4**    | 1.41 | 2.29 | 1.70 | 1.50 |
| **5**    | 2.30 | 1.58 | 1.82 | 1.78 |
| **6**    | 1.67 | 1.84 | 1.78 | 1.55 |
| **7**    | 1.42 | 1.61 | 1.42 | 1.35 |
| **8**    | 2.05 | 2.35 | 2.22 | 1.98 |
| **9**    | 1.64 | 1.94 | 1.78 | 1.47 |
| **Avg.** | 1.68 | 1.83 | 1.72 | 1.59 |

**Table 10.** AF4 decoder results for synthetic encoding, where only one point is activated.

| | Code | | | | Output |
|------|---|---|---|---|---|
| **#1** | ▪ | | | | ⁊ |
| **#2** | | ▪ | | | ╎ |
| **#3** | | | ▪ | | ━ |
| **#4** | | | | ▪ | ⁄ |

**Table 11.** The sums of occurrence intensities (values in the encoding) for each visual feature AF4. It must be emphasized that the results in this table are relative, so they can only be compared to the numbers from the same table.

|      | #1   | #2   | #3   | #4   |
|------|------|------|------|------|
| **0**    | 4.89 | 5.12 | 3.20 | 5.25 |
| **1**    | 3.90 | 4.19 | 2.14 | 3.63 |
| **2**    | 5.07 | 5.20 | 3.36 | 5.77 |
| **3**    | 5.18 | 5.44 | 3.58 | 5.08 |
| **4**    | 5.03 | 5.72 | 2.89 | 5.07 |
| **5**    | 4.92 | 4.97 | 3.26 | 4.76 |
| **6**    | 5.38 | 5.00 | 3.17 | 4.92 |
| **7**    | 4.78 | 5.14 | 2.88 | 4.33 |
| **8**    | 5.41 | 5.72 | 3.46 | 5.27 |
| **9**    | 5.08 | 5.00 | 3.21 | 4.65 |
| **Avg.** | 4.95 | 5.14 | 3.10 | 4.85 |

**Table 12.** `AF5` decoder results for synthetic encoding, where only one point is activated.

| | Code | | | | | Output |
|---|---|---|---|---|---|---|
| **#1** | ▪ | | | | | ↘ |
| **#2** | | ▪ | | | | ➴ |
| **#3** | | | ▪ | | | ❙ |
| **#4** | | | | ▪ | | ➖ |
| **#5** | | | | | ▪ | ✦ |

**Table 13.** The sums of occurrence intensities (values in the encoding) for each visual feature `AF5`. It must be emphasized that the results in this table are relative, so they can only be compared to the numbers from the same table.

| | #1 | #2 | #3 | #4 | #5 |
|---|---|---|---|---|---|
| **0** | 2.48 | 2.71 | 2.81 | 3.68 | 3.03 |
| **1** | 1.38 | 1.25 | 1.47 | 2.21 | 1.57 |
| **2** | 2.73 | 3.18 | 2.59 | 3.87 | 3.11 |
| **3** | 2.73 | 3.38 | 2.32 | 4.02 | 2.88 |
| **4** | 2.11 | 2.34 | 2.27 | 3.55 | 2.54 |
| **5** | 2.61 | 2.94 | 2.09 | 4.10 | 2.72 |
| **6** | 2.50 | 2.79 | 2.50 | 3.64 | 2.87 |
| **7** | 1.89 | 2.56 | 2.09 | 3.43 | 2.40 |
| **8** | 2.85 | 3.26 | 2.64 | 3.76 | 2.96 |
| **9** | 2.18 | 2.85 | 2.16 | 3.45 | 2.55 |
| **Avg.** | 2.33 | 2.70 | 2.28 | 3.55 | 2.65 |

The conclusions drawn from Tables 9, 11, and 13 can be further explained with proper illustrations. Decoding a single point did not provide a satisfactory understanding of the visual features (as was the case with feature #3 in `AF5`), which provides motivation to search for a better way of feature visualization. Instead, we can use the decoder for the actual multiple-pixel combinations generated for inputs from the test set. The sparse representation can be further split into separate matrices, related to different visual features. The decoding of a single matrix can be considered as partial reconstruction, which consists only of occurrences of the corresponding feature. For example, using only the first channel of the model with manually selected features will result in a partial reconstruction of a digit that consists of horizontal lines only. Additionally, in order to explain the limitations of presented methods, specific digits with especially low and especially high autoencoder errors were chosen for this visualization. Selected results for the discussed models are presented in Figures 8–10.

The rows described as encoding in Figures 8–13 contain the visualizations of tuples of sparse matrices. For the sake of readability, the active elements, which are naturally rare, were magnified threefold. The features row provides information on the distribution of particular types of visual features on the image plane. This may be associated with a particular digit segment that possesses that feature. It must be emphasized, however, that the decoder output cannot be described as a sum of separate features—the CNN-based decoder function is nonlinear and non-additive. The encoded

information on the presence of a selected feature indicates not only the presence of specific digit segments associated with that feature, but it may also be indicative of the absence of other features, as is the case with digit 8 (Figure 8).

A similar approach was applied to larger images. In order to include even more information in the presented visualization, images containing both digits and other symbols were used. The results are shown in Figures 11–13. The autoencoders were trained in a digit-specific way, but the test images used in this case contain other symbols as well. The selected non-digit shapes, however, generated visible errors—some segments were erroneously enlarged, merged, broken into pieces or blurred.

An additional demonstration of the proposed method using a longer text fragment, which is a 512 × 128 scan of a postal address, is presented in Figures 14–16. They depict the encoding contents and decoder outputs for the three proposed models. The number of active elements in the sparse encoding is proportional to the image size, which is visibly larger than in the other examples. Digits are clearly readable in the decoder output, as their size is roughly similar to the MNIST samples. Some of the most significant errors occur for the pairs of letters that are especially close to each other, which are visible in the second line of text.

**Figure 8.** Sample encoding of selected digits performed by MF4 model. Apart from the input and output data and highly-sparse encoding (up to 5 non-zero elements in each encoding matrix), visualizations of single features are presented. Each visualization was acquired by decoding a synthetic code, where one of the visualized matrices was used, and all the other matrices were filled with zeros.

| Model | AF4 | | | |
|---|---|---|---|---|
| Input | 0 | | | |
| Encoding | | | | |
| Features | | | | |
| Output | 0 | | | |
| Input | 1 | | | |
| Encoding | | | | |
| Features | | | | |
| Output | 1 | | | |
| Input | 8 | | | |
| Encoding | | | | |
| Features | | | | |
| Output | 8 | | | |

**Figure 9.** Sample encoding of selected digits performed by `AF4` model. Highly sparse encoding (up to 5 non-zero elements in each encoding matrix) was used. Feature-specific decodings are presented, similar to those in Figure 8.

| Model | AF5 | | | | |
|---|---|---|---|---|---|
| Input | 0 | | | | |
| Encoding | | | | | |
| Features | | | | | |
| Output | 0 | | | | |
| Input | 1 | | | | |
| Encoding | | | | | |
| Features | | | | | |
| Output | 1 | | | | |
| Input | 8 | | | | |
| Encoding | | | | | |
| Features | | | | | |
| Output | 8 | | | | |

**Figure 10.** Sample encoding of selected digits performed by `AF5` model. Highly sparse encoding (up to 5 non-zero elements in each encoding matrix) was used. Feature-specific decodings are presented, similar to those in Figure 8.

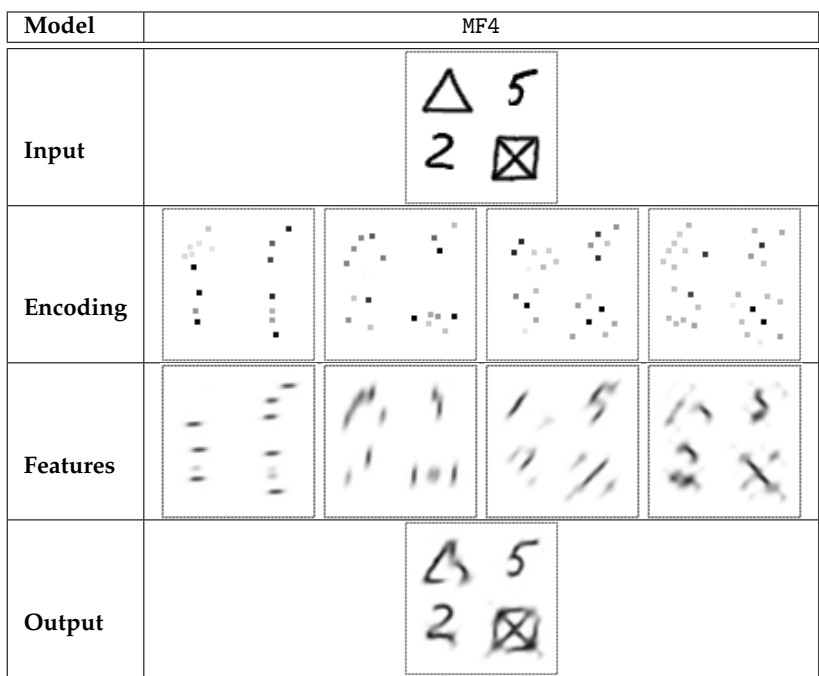

**Figure 11.** Sample encoding of a larger image that contains both digits and other symbols, performed with the MF4 model. The presented feature-specific decodings were generated in the same way as those presented in Figure 8.

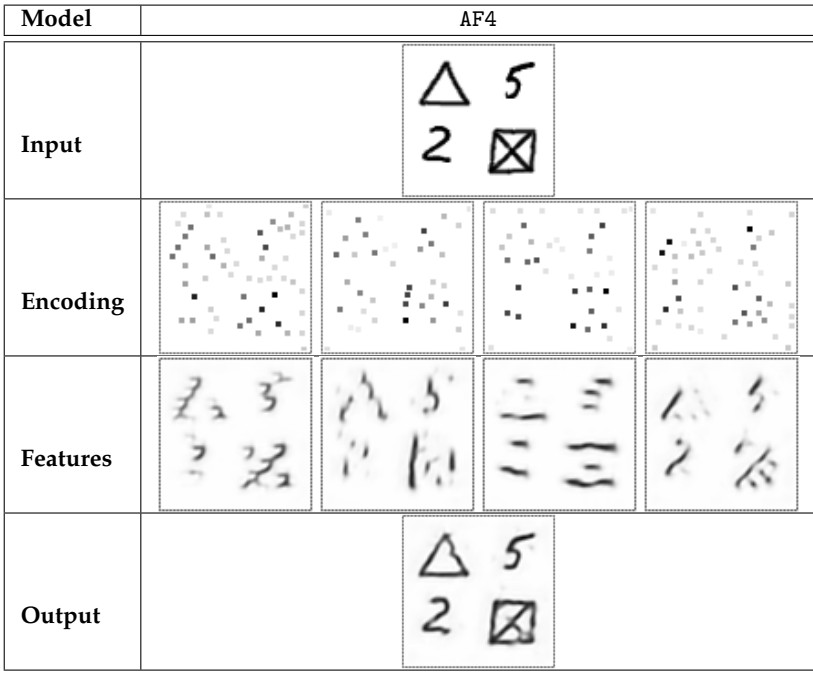

**Figure 12.** Sample encoding of a larger image that contains both digits and other symbols, performed with the AF4 model. The presented feature-specific decodings were generated in the same way as those presented in Figure 8.

| Model | AF5 |
|-------|-----|
| Input |  |
| Encoding |  |
| Features |  |
| Output |  |

**Figure 13.** Sample encoding of a larger image that contains both digits and other symbols, performed with the AF5 model. The presented feature-specific decodings were generated in the same way as those shown in Figure 8.

| Model | MF4 |
|-------|-----|
| Input |  |
| Encoding |  |
| Features |  |
| Output |  |

**Figure 14.** Sample encoding of a postal address scan that contains both digits and letters, performed with the MF4 model. The presented feature-specific decodings were generated in the same way as those shown in Figure 8.

| Model | AF4 |
|-------|-----|
| Input | INSTITUTE OF INFORMATION TECHNOLOGY<br>LODZ UNIVERSITY OF TECHNOLOGY<br>WOLCZANSKA 215, 90-924 LODZ, POLAND |
| Encoding | |
| Features | |
| Output | INSTITUTE OF INFORMATION TECHNOLOGY<br>LODZ UNIVERSITY OF TECHNOLOGY<br>WOLCZANSKA 215, 90-924 LODZ, POLAND |

**Figure 15.** Sample encoding of a postal address scan that contains both digits and letters, performed with the AF4 model. The presented feature-specific decodings were generated in the same way as those shown in Figure 8.

| Model | AF5 |
|-------|-----|
| Input | INSTITUTE OF INFORMATION TECHNOLOGY<br>LODZ UNIVERSITY OF TECHNOLOGY<br>WOLCZANSKA 215, 90-924 LODZ, POLAND |
| Encoding | |
| Features | |
| Output | INSTITUTE OF INFORMATION TECHNOLOGY<br>LODZ UNIVERSITY OF TECHNOLOGY<br>WOLCZANSKA 215, 90-924 LODZ, POLAND |

**Figure 16.** Sample encoding of a postal address scan that contains both digits and letters, performed with the AF5 model. The presented feature-specific decodings were generated in the same way as those shown in Figure 8.

The presented illustrations involved 28 × 28 MNIST samples, 80 × 80 images with multiple symbols and 512 × 128 scan of a postal address. However, the method is scalable for any size of an input image. The execution time is proportional to the number of pixels, as the relations presented in the Figure 17 are roughly quadratic. The presented calculation time is very small, despite the standard GPU set-up being used for a single image. Processing multiple images of similar size in batches would reduce the average per-image processing time even further.

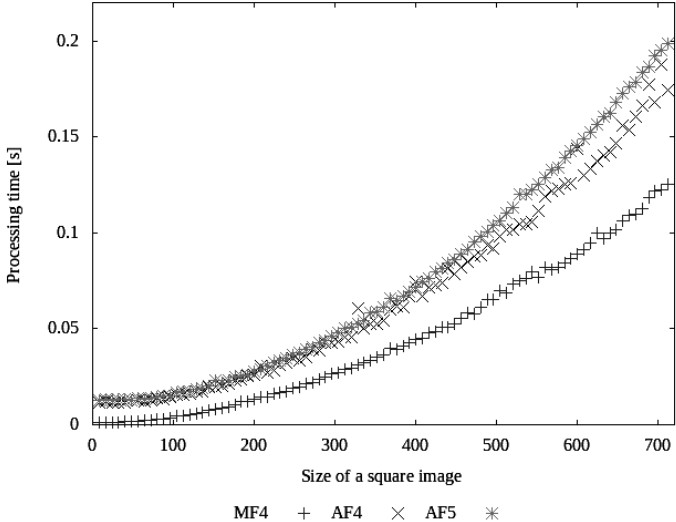

**Figure 17.** Processing times of MF4, AF4, and AF5 models for square images of different sizes. As in the case of the standard training and test process, these results were achieved using GPU.

## 6. Conclusions

This paper has presented a novel method of image content representation. In our approach, we propose to encode an image as a tuple of sparse matrices that describes the intensity and position of selected visual features occurring in the image. The method was validated through a series of experiments on the MNIST data set [7]. The presented simple variant, in which each matrix was reduced to local maxima, provided the ability to generate sparse matrices where no more than 6.25% of elements were preserved. However, as revealed by further analysis, very sparse matrices with no more than five elements preserved in each 28 × 28 matrix (which is less than 0.64% of elements preserved) were sufficient to keep a low autoencoder error rate and obtain a classifier accuracy of 98.40%.

The application of the method to the classification task provided satisfactory results, outperforming the classical sparse coding solutions [23,25]. In our approach, the method of encoding was the same, regardless of the image class, because no class-dependent information was used in the training process. Thus, the presented method can be considered unsupervised. This is a relevant factor that has to be taken into account when comparing the results of the present study with those in [22]—our results are not as good as those of the supervised variant presented in that work but outperform those of the unsupervised variant.

The presented classifier also performs better than most solutions based on spiking neural networks [28,29]. It is not as good as some models presented in [31], but it must be emphasized that the classification accuracy comparison is not the key aspect in this particular case. Spiking neural networks possess different properties, such as the ability to handle noise [29]. Some of the models, such as LIF neurons [27,28] or STDP learning [29], suffer from too high complexity, or unsatisfactory sparsity level of the generated representation (especially in [31]).

The sparse representation based on visual features made it possible to perform a detailed analysis of the nature of the selected features. We presented both context-free feature visualizations

(see Tables 8, 10, and 12) and digit-specific distribution of particular features (see Figures 8–13). Additionally, per-digit statistics of feature occurrences were discussed (see Tables 9, 11, and 13).

The presented method is based solely on convolutional layers and point-wise operations (activation functions). This makes the feature detection invariant to translation. Consequently, the pretrained autoencoder can process the input images of any size, which was demonstrated on the basis of a data set with larger images, each containing multiple digits (see Section 4.4). Experiments using non-digit symbols were also performed (see Figures 11–13), in order to show that the generated features were adjusted to the selected data set—not surprisingly, the autoencoder trained in the proposed way performed markedly worse on non-digit characters.

The study was based on the MNIST data set [7], which is relatively easy to analyze. However, further tests are needed to verify the applicability of the method to more complex databases, such as the ImageNet data set [37] or data from the Pascal Visual Object Classes Challenge [38]. In order to be applied to images of real-life objects, the method would have to be enhanced with the ability to recognize more complex and more numerous visual features.

The different visual features used in the presented approach are sensitive to object rotation and size. In particular, the images from the MNIST data set were easy to describe with lines, and the rotation of the line was the key element that identified the features. However, the task of feature detection in different rotations can be approached in multiple ways. Recent works on CNNs have provided new advances to transformation-invariant CNNs [14,39,40]. The prospect of combining those methods with the proposed representation, resulting in additional information about orientation and scale of detected keypoints, is another promising option worth pursuing for further development.

**Author Contributions:** Conceptualization, A.T., P.T., and B.S.; methodology, P.T. and A.T.; software, P.T.; validation, P.T.; formal analysis, P.T.; investigation, P.T., A.T., and B.S.; resources, P.T. and A.T.; data curation, P.T.; writing—original draft preparation, P.T, A.T., and B.S.; writing—review and editing, P.T., A.T., and B.S.; visualization, P.T.; supervision, A.T.; project administration, A.T.; funding acquisition, A.T. All authors have read and agreed to the published version of the manuscript.

**Funding:** This project has been partly funded with support from the National Science Centre, Republic of Poland, Decision Number DEC-2012/05/D/ST6/03091.

**Conflicts of Interest:** The authors declare no conflicts of interest.

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
