# Peer review of "Automatic Identification of Local Features Representing Image Content with the Use of Convolutional Neural Networks"

_applsci, doi:10.3390/app10155186_

Round 1

Reviewer 1 Report

Pros: In this manuscript authors propose a novel technique of image analysis. In this technique a set of images are reduced to individual sparse matrices which show a bunch of features that eventually provide information about the intensity at a particular location. Authors compare their work with other state of the art neural network models to show performance benefits.

Cons:

The paper comes with its limitations, the key issue the author feels is that the paper lacks new research outputs. The reviewer feels, the analysis the authors have done for the images is perhaps exhaustively researched mainly shown in the number of papers that have been cited. The key question therefore, is what are the main research contributions that could be used to further the CNN research?

The development of a CNN module is not completely new. Authors have done very good work in terms of developing this, but again, the question comes back to the complexity of the model. The main ingredient that the authors propose to use the intermediate step of sparse matrix building, this is very much like what is done in some of the recent works in Reinforcement learning such as "V. Balasubramanian, M. Aloqaily, O. Tunde-Onadele, Z. Yang and M. Reisslein, "Reinforcing Cloud Environments via Index Policy for Bursty Workloads," NOMS 2020 - 2020 IEEE/IFIP Network Operations and Management Symposium, Budapest, Hungary, 2020, pp. 1-7, doi: 10.1109/NOMS47738.2020.9110417". Reviewer would like to know, if the simple modelling of an agent such as this one, would help reduce the complexity of building a sparse matrix? It is recommended that authors provide some difference between the methods to showcase their novelty of model.

English Corrections and Readability:

(1) There are many typos and sentence formations which need to be corrected mainly 

-->page 5, line 201, "the propose in this work"..what do the authors want to say here?

--> page 16, line 462 " Another remarkable work on sparse...in the SRC"-->".. is the SRC"

In this way, there are plenty of corrections that the authors are recommended to revise. 

Author Response

July 18, 2020

Reviewer of Applied Sciences journal

Dear Sir or Madam,

At the beginning we would like to express our gratitude for your valuable remarks. Following your comments and suggestions of the other reviewers we have made some modifications in our paper. All those modifications were highlighted.

Detailed response:

The paper comes with its limitations, the key issue the author feels is that the paper lacks new research outputs. The reviewer feels, the analysis the authors have done for the images is perhaps exhaustively researched mainly shown in the number of papers that have been cited. The key question therefore, is what are the main research contributions that could be used to further the CNN research?

We have decided to use MNIST data set due to its popularity. It was our conscious decision and there are two reasons behind it. First of all, many researchers know this database. Secondly, it contains small images with precisely defined objects which are well known for everybody, not only for the image processing community. Consequently, all the readers should have some intuition concerning these images and thus the presented results should be easier to interpret. This interpretability was one of our main design goals. We believe, after literature review, that our approach to automatic identification of interpretable, spatially located features is new and in its first publication it is reasonable to use a simpler data set to clearly present its working principles.

To better show the properties of the described method and, by the way, to present possible application of representation generated from MNIST database, we have added Figures 14, 15 and 16. We have applied an encoder part of the trained autoencoder to larger images containing many objects of similar characteristic (in this case this is a scan of a handwritten postal address). Analyzing the spatial distribution of the features (points) in the encoding matrices, we can construct a tool for identification and location of digits. This task is currently under investigation.

The development of a CNN module is not completely new. Authors have done very good work in terms of developing this, but again, the question comes back to the complexity of the model. The main ingredient that the authors propose to use the intermediate step of sparse matrix building, this is very much like what is done in some of the recent works in Reinforcement learning such as "V. Balasubramanian, M. Aloqaily, O. Tunde-Onadele, Z. Yang and M. Reisslein, "Reinforcing Cloud Environments via Index Policy for Bursty Workloads," NOMS 2020 - 2020 IEEE/IFIP Network Operations and Management Symposium, Budapest, Hungary, 2020, pp. 1-7, doi: 10.1109/NOMS47738.2020.9110417". Reviewer would like to know, if the simple modelling of an agent such as this one, would help reduce the complexity of building a sparse matrix? It is recommended that authors provide some difference between the methods to showcase their novelty of model.

The mentioned work is far from our areas of expertise and thus we are not sure to which part of this paper we should refer to. However, we are - of course - aware of the fact that building of sparse matrices is an important element of many different practical tasks (e.g. more efficient storage and computations). In our work we generate sparse matrices using convolutional autoencoder. They are expected to preserve the information contained in the original matrices (images). We use matrices, and not vectors, because not only do we want to compress the information, but we want to extract the information about localization of the visual building blocks as well. This is a very specific application which requires matrices only because images are represented as regular grids of pixels. Nevertheless, the proposed approach is general. CNNs are designed to work with matrix-like structures but interpretation of these structures is irrelevant. If an autoencoder is used, one can obtain an encoder which generates sparse matrices preserving the whole information. In order to use the resulting sparse matrices, another, dedicated tool unit must be designed (an example is the classifier described in Section 4.3). An alternative solution, which is not presented in our work, could be to train a CNN directly performing a specific task (e.g. classification) with enforced sparsity inside. In this case, however, the sparse information would not preserve the whole information about the input, but only this part which is required to accomplish a given goal. To conclude, we really appreciate this remark as it has shown a wide field of applications of our approach. We have added this information in Section 2.4.

English Corrections and Readability:

(1) There are many typos and sentence formations which need to be corrected mainly

-->page 5, line 201, "the propose in this work"..what do the authors want to say here?

--> page 16, line 462 " Another remarkable work on sparse...in the SRC"-->".. is the SRC"

In this way, there are plenty of corrections that the authors are recommended to revise.

We have corrected the first typo. In the second case everything is fine. In this case, the authors of the cited work were maximizing the margin between classes in a classification task where sparse representation was used. SRC refers to sparse representation-based classification (this acronym was introduced in the cited publication). We have also revised our paper carefully to improve its readability and to remove similar mistakes.

Yours faithfully,

Authors

Reviewer 2 Report

This paper presents novel local features to reduce the representation of image content and its applications based on CNNs.
The procedure of the proposed method and the contributions of this paper are clearly claimed.
The characterizing performance of the proposed method is verified via simple experimental examples and concrete discussions.
All figures and tables are clear.
I could not find the problems of this paper on both novelty and practicality.
Thus, I think this paper is well-written and acceptable.

Author Response

July 18, 2020

Reviewer of Applied Sciences journal

Dear Sir or Madam,

At the beginning we would like to express our gratitude for your valuable remarks. We are glad that our contribution was clearly claimed.

Following suggestions of the other reviewers we have made some modifications in our paper. All these modifications were highlighted.

Yours faithfully,

Authors

Reviewer 3 Report

Authors have presented a feature based study using convolutions neural networks. Although it is not highly novel, it’s of some importance. However, authors have missed out certain results which I would recommend the authors to conduct to make it a more robust study. 

Authors should the class activation mapping results of CNNs to completely understand its output. Here is a reference which works on the same for medical imaging.

https://arxiv.org/abs/1912.09621

Authors present the results using feature reduction algorithms. I would recommend them to study these using PCA and Auto encoders as mentioned in the following papers for their complete study.

PCA

https://ieeexplore.ieee.org/abstract/document/7856826/

Author Response

July 18, 2020

Reviewer of Applied Sciences journal

Dear Sir or Madam,

At the beginning we would like to express our gratitude for your valuable remarks. Following your comments and suggestions of the other reviewers we have made some modifications in our paper. All these modifications were highlighted.

Detailed response:

Authors should the class activation mapping results of CNNs to completely understand its output. Here is a reference which works on the same for medical imaging. https://arxiv.org/abs/1912.09621

The mentioned work uses the class activation mapping (CAM) method described in https://arxiv.org/abs/1512.04150. This method, among others, can be used for identification and visualization of discriminative image regions as well as for weakly-supervised object localization. In that approach, a CNN network is trained to classify images but typical fully-connected layers are replaced by global average pooling (GAP) followed by one fully-connected soft-max layer. The CAM for a given class can be found as linear combination of final feature maps generated by convolutional layers with weights corresponding to a specific network output. Since the size of the class activation map is equal to the size of the final feature maps, it must be up-sampled to be comparable with the input image.

In our work we look for building blocks of a given class of images in a totally unsupervised way (convolutional autoencoder) so CAM cannot be used directly here. We have conducted an image classification experiment in Section 4.3 only to prove that reduced representation can be effectively used in some practical tasks (or, in other words, to prove that the required information about image content is still preserved in sparse matrices). This reduced representation, and not the original image, was the input of the classifier (4 or 5 channels containing only a few pixels with non-zero value). Generating CAM for such images would simply lead to identification of the previously located characteristic points. Moreover, our classifier uses traditional fully-connected layers and not GAP with fully-connected soft-max layer.

The CAM technique, however, is interesting from another point of view. It can be considered as a related work which is also devoted to CNN-based semantic interpretation of image content. Authors of that work have shown that CAM can be of use for identification of class-specific units. There are of course essential differences in comparison to our approach. To mention only two of them, our method is fully unsupervised and it allows for a precise localization of semantic blocks (CAM activation maps are rather blurred due to the required up-sampling). Information about CAM was added to our work in Section 2.2.

Authors present the results using feature reduction algorithms. I would recommend them to study these using PCA and Auto encoders as mentioned in the following papers for their complete study.

PCA

https://ieeexplore.ieee.org/abstract/document/7856826/

We indeed present a feature reduction algorithm. There is, however, one crucial difference between our approach and the principal component analysis (PCA). PCA treats its input and output as feature vectors where no spatial relationship between their elements is taken into account. Consequently, we could treat input images as vectors and reduce dimensionality of such feature space, but we could not interpret the resulting features as visual building blocks. Typical non-convolutional autoencoders do the similar thing but, in contrast to PCA, they can also construct non-linear mapping between the original and the reduced features.

The main goal of our research was to reduce image representation leading to features that can be spatially localized or, in other words, we want to obtain features that represent visual building blocks. That is why we use convolutional autoencoder. It takes the spatial relationships between the reduced features and, which is its additional advantage, it performs feature reduction in a local way. The resulting features are calculated only on the basis of features belonging to a respective receptive field. Such weight sharing, typical for CNNs, reduces the number of the trainable parameters and, in our case, allows to detect the same features in different places of the image plane. The difference presented above was added to our work in Section 2.4.

Yours faithfully,

Authors

Reviewer 4 Report

The idea of the paper is to apply deep learning techniques to extract features from images in order to improve their classification. This idea has been deeply explored in the literature, so it is difficult to evaluate the novelty of the paper. Also there are several strong concerns about this paper that the authors need to solve before considering it for publication:

- The evaluation is poor and confusing. The authors are not evaluating with other techniques under similar circumstances. During section 4.3, you should include more techniques under the same experimental setup, perform a proper parameter tuning with every technique and then create a table like table 6 comparing the accuracy results. Also, this experiment must run multiple times, and they need to include a statistical test (such as the Wilcoxon test), during the comparison. Otherwise, the contribution of the tool is unclear with respect to the state-of-the-art.

- The evaluation lacks a ground truth, which is obvious in Table 5. It is difficult to understand whether those results are good or not apart for the comparison among the techniques introduced by the authors. The main goal should be to prove that this method can actually maintain accuracy reducing the amount of time and memory used for that. However, that kind of evaluation is not part of this paper. In order to consider the paper for publication, the authors should run a proper comparison as the mentioned before.

- The two datasets selected have very small images MNIST have 28x28 images and the augmented one has 80x80. This suggests that there is a big problem with the scalability of the technique, which is not properly studied in this work. The authors should conduct a scalability study to measure time and memory consumption of each approach and compare it with other techniques from the state of the art. They also have to use more complex images.

- Section 4.4 presents an unclear experiment. What is the goal of mixing the images? It looks like the authors aim to find the specific digits into more complex images by applying their technique, but this is not the goal presented in the paper. I recommend the authors to apply their technique to the complex images in order to extract which digits they identify into each image with their classifier.

- About the methodologies, there is no discussion about the predefined parameters for AF 4, AF 5 and MF 4, why are you choosing these specific configurations and not others? Is it possible that MF5 or AF 6 would be better?

- There is a methodology problem with the feature extraction process: it looks like the authors extract the features from the whole dataset and then they divide the data into train and test (section 4.3), however, the test data can not be used for the feature extraction process, it must be fresh data for the whole classification pipeline. Please, correct this.

- Line 454 to 492 used the results of other techniques as a comparison, but this comparison is bias, as they are not in exactly the same setup. Authors must run the experiments on these tools using exactly the same train and test data and performing a proper automatic selection of parameters for each technique.

Author Response

July 18, 2020

Reviewer of Applied Sciences journal

Dear Sir or Madam,

At the beginning we would like to express our gratitude for your valuable remarks. Following your comments and suggestions of the other reviewers we have made some modifications in our paper. All these modifications were highlighted.

Detailed response:

The idea of the paper is to apply deep learning techniques to extract features from images in order to improve their classification. This idea has been deeply explored in the literature, so it is difficult to evaluate the novelty of the paper.

The main goal of the paper was to force a convolutional neural network to automatically find highly reduced and interpretable set of localized features which would contain almost whole information about the image content. An additional objective, which was accomplished, so to say, by the way while using the CNN autoencoder, was the ability to connect these features with visual building blocks composing the whole image (they can be found using the decoder part of the autoencoder). The motivation for searching for such representations was presented in Section 1. In other words, our goal was not to extract features from images to improve their classification. Indeed, we have conducted an image classification experiment in Section 4.3. However, we have done it only to prove that the reduced representation can be effectively used in some practical tasks (or, in other words, to prove that the required information about the image content is still preserved in sparse matrices). This misunderstanding had an influence on some other comments contained in the review. We have revised our paper carefully to improve its readability (some sentences were rephrased) and we have tried to address all the problems indicated by the reviewer (discussed further). We believe that now our work will be less confusing for the reader.

- The evaluation is poor and confusing. The authors are not evaluating with other techniques under similar circumstances. During section 4.3, you should include more techniques under the same experimental setup, perform a proper parameter tuning with every technique and then create a table like table 6 comparing the accuracy results. Also, this experiment must run multiple times, and they need to include a statistical test (such as the Wilcoxon test), during the comparison. Otherwise, the contribution of the tool is unclear with respect to the state-of-the-art.

In order to improve the clarity of both the objectives and the results of our evaluation, we propose an extended explanation in Section 4.1. The closest possible comparisons with other techniques were discussed in Section 4.3. However, for each of the discussed citations, we believe that the differences in details should not be overlooked. These include the size of a training subset [23], the structure [25] or the size [28,31] of the sparse representation and the application of the sample labels [22]. Therefore, placing these results in the same Table 6 would be misleading, and we attempt to clarify our position with respect to the state-of-art in the suitable paragraph, where complete explanation is provided.

It needs to be emphasized that the image classification task was utilized as one of the tools for analysis of the encoders, but it is not the key objective of the proposed paper. We use the results of decoders and classifiers trained in fixed conditions to compare the experimental setups, but we do not compete in attempt to provide globally optimal decoders or classifiers, as such a pursuit would significantly shift the focus of this paper.

The results of autoencoders training were roughly repeatable in terms of global decoding errors, but the permutation of features is attributed to a chance with relation to the initial training weights. Analysis based on multiple training instances would extend our paper in yet another way, but we believe that the acquired encoders are applicable to perform the thorough analysis presented in further sections.

- The evaluation lacks a ground truth, which is obvious in Table 5. It is difficult to understand whether those results aregood or not apart for the comparison among the techniques introduced by the authors. The main goal should be to prove that this method can actually maintain accuracy reducing the amount of time and memory used for that. However, that kind of evaluation is not part of this paper. In order to consider the paper for publication, the authors should run a proper comparison as the mentioned before.

There exists no external comparison to the results from Table 5. The table presents per-subset averages of total quadratic errors. The error is always nonnegative, and the expected quadratic difference between uniform random matrices is 130.67. Our models can be at very least considered good with regard to this range, as even the highest average error for a specific subset is lower than 15.0. For technical reasons, there is no literal ground truth or fully adequate literature reference. The error of 0.0 is pursued (we expect output to be the same as input), but we can presume that it is not possible given the limitations on the sparse representation. Since MF4 model is designed manually and it is remarkably the simplest one amongst the considered setups, it can be considered, to some extent, as a reference point for the other setups.

We believe that the amounts of time and memory required for the computations with the proposed models do not pose an issue. We have included additional information on the model size – each encoder is described by less than 30 000 numbers. Additional measurement of the processing time for a series of larger images was added to the paper as well – the time is proportional to the image size, and even the out-of-context propagation for 700x700 images takes less than 0.2s for each of the setups (Figure 17).

Reduction of the encoder model complexity is indeed possible, up to the point where no useful visual features are identified. Our trials have shown that a significant reduction, such as using twice less filters, leads to such a result. On the other hand, the proposed setup provided us with useful features within the limitations on sparse representation, exceeding the expectations set by the human design, as AF4 performed better than MF4. Based on these observations, we believe that our decision regarding resources usage was suitable for the purposes of this paper. We made sure to clarify this, as the explanation of our experiments was extended in Section 4.1.

- The two datasets selected have very small images MNIST have 28x28 images and the augmented one has 80x80. This suggests that there is a big problem with the scalability of the technique, which is not properly studied in this work. The authors should conduct a scalability study to measure time and memory consumption of each approach and compare it with other techniques from the state of the art. They also have to use more complex images.

As the encoder and decoder parts of the proposed neural networks are fully convolutional, literally any image can be processed with our technique. The demonstration of the scalability and applicability for more complex images was added to the Section 5 (Figures 14, 15 and 16). The processing time is proportional to the image size, which indicates the proper scalability of our method (Figure 17). The decoding results for an example piece of text is erroneous for letters and sequences of letters, as the visual features were computed specifically for the MNIST dataset. However, the digits are readable regardless of the context.

- Section 4.4 presents an unclear experiment. What is the goal of mixing the images? It looks like the authors aim to find the specific digits into more complex images by applying their technique, but this is not the goal presented in the paper. I recommend the authors to apply their technique to the complex images in order to extract which digits they identify into each image with their classifier.

We believe that the experiment presented in Section 4.4 is remarkably important with respect to the objective of our paper. The proposed sparse representation is related to the visual features that are deliberately smaller than the whole digit, and as such – we do not identify the whole digits. The discussed experiment allowed us to demonstrate that processing digits with our model works for images with broader context as well. What is more, additional experiments made it possible to visualize the potential and limitations of our models for other symbols, which was explored in Section 5. In order to address this apparent ambiguity, we have also extended the explanation in Section 4.4 with this issue in mind.

- About the methodologies, there is no discussion about the predefined parameters for AF 4, AF 5 and MF 4, why are you choosing these specific configurations and not others? Is it possible that MF5 or AF 6 would be better?

We believe that MF4 is special, because of how natural it is and how well it fits the limitations of a filter matrix – as it is shown in Table 3. The theoretical possibility of any other set of manually designed filters, such as MF5, would be ambiguous and require approximation. AF4 and AF5 were selected, respectively, as a direct comparison to MF4 and as the most natural next step. Further extensions, such as AF6 are possible – and we believe that it would be the right approach for data sets more complex than MNIST. However, low number of features makes the visual analysis easier, as each feature can be distinguished with just a glance on the appropriate diagram. In order to make this issue clear in the paper, we have extended Section 4.1.

- There is a methodology problem with the feature extraction process: it looks like the authors extract the features from the whole dataset and then they divide the data into train and test (section 4.3), however, the test data can not be used for the feature extraction process, it must be fresh data for the whole classification pipeline. Please, correct this.

We are glad to assure you that the mentioned problem was avoided, as the autoencoder models were trained only with the training portion of the MNIST data set. The evaluation of decoders was based on the test part of MNIST, so our results can be considered general in terms of this data set, with no risk of simple data memorization affecting the results. We have made sure to clearly highlight this fact in Section 4.1.

- Line 454 to 492 used the results of other techniques as a comparison, but this comparison is bias, as they are not in exactly the same setup. Authors must run the experiments on these tools using exactly the same train and test data and performing a proper automatic selection of parameters for each technique.

The nature of the presented comparison was explained in Section 4.3. The issue of comparison is not limited to the selection of the training data set – this specific aspect is mentioned only with regard to one of the related works [23]. In most of the cases, we use the same MNIST data set as the related works. However, the experimental circumstances are different not only because of the technical setups, but because of diverse specific objectives of the considered methods as well. The sources of bias are discussed in the paper, and we have extended Section 4.3 to make them clearer.

Yours faithfully,

Authors

Round 2

Reviewer 3 Report

Authors have addressed most of my concerns. However, they could add the  references (mentioned in the previous review) in the edited sections so that reader could understand the difference. 

Reviewer 4 Report

The authors have addressed all my comments and the paper is now ready for publication